



**Occurrence of discontinuities in the ozone concentration data from three reanalyses**
**Peter Krizan, Michal Kozubek, Jan Lastovicka**
*Institute of Atmospheric Physics, Czech Academy of Sciences, 14100 Prague, Czech Republic;*
**Correspondence:** Peter Krizan (krizan@ufa.cas.cz)
**Abstract**
*Ozone is a very important trace gas in the stratosphere and thus we need to know its temporal*
*evolution over the globe. The ground based measurements are rare, especially in the Southern*
*Hemisphere. Satellite ozone data have broader coverage, but they are not available from*
*everywhere. On the other hand, the reanalyse data have regular spatial and temporal*
*structure, which is very good for trend analyses. But there are discontinuities in these data.*
*These discontinuities may influence the results of trend studies. The aim of this paper is to*
*detect the discontinuity occurrence (DO) in the following reanalyses: MERRA-2, ERA5 and*
*JRA-55 with the help of the Pettitt homogeneity test at all common layers above 500 hPa. The*
*discontinuities are sorted according to their size to the significant and the insignificant ones;*
*the former can affect the ozone trend studies. It was shown that DO for the significant*
*discontinuities is the smallest in JRA-55. In the upper model layers, the discontinuity*
*occurrence is the highest. The other area of high DO is the troposphere.*
**1. Introduction**
Ozone is an important trace gas in the atmosphere, because it protects the biota of the Earth
from the harmful ultraviolet radiation.  In the beginning of the 1970s the ozone research was
mostly interested in the connection between the ozone layer and the supersonic transport. The
great challenge was the discovering of the Antarctic ozone hole. The origin of the ozone hole
is in the chemical reactions of anthropogenic halogen radicals, which destroy ozone (Solomon,
1999). As a consequence of these results, the Montreal Protocol and its amendments was signed,
which stopped the production of the ozone-depleting substances (ODS). This protocol led to a
decrease of ODS concentrations in the stratosphere (WMO, 2014). There are some signs of the
ozone layer recovery, especially in the upper stratosphere (Harris et al., 2015). In addition, all
models predict the future recovery of the ozone layer (Eyring et al., 2010). The concentration of
ODS is not the only factor that has impact on the ozone layer. In addition, the greenhouse
cooling of the stratosphere (Waugh et al., 2009) and changes in the Brewer–Dobson circulation
influence the ozone concentration. In such situation, proper trend analysis is necessary for the
understanding of the ozone layer behaviour. Trend analyses based on ground-based data (Mc
Landress et al., 2009; Krzyscin et al., 2008) suffer with data being measured at single locations
only, and the number of ground-based stations is insufficient, especially in the Southern
Hemisphere. Satellite ozone data have broader coverage, and these data are widely used in trend
analysis in the ozone research (e.g., Jones et al., 2009).  In order to have long data series of
satellite ozone measurements, composites from various satellites have been used (e.g., Ball et
al., 2019; Bourassa et al., 2014; Tummon et al., 2014). But in some areas, it is impossible to
measure ozone (polar night, below dense clouds) by satellite. SPARC Report No. 9 (2019)
critically summarizes the current state-of-the-art as concerns trends in stratospheric ozone.





On the other hand, the ozone data from reanalysis are generated in equal time step and they
are spatially homogeneous, but there is a big question of the suitability of these data for trend
analysis due to the occurrence of discontinuities (Bengtsson et al., 2004). They can be caused
by satellite or instrument replacements or by the assimilation of not homogenous basic
parameters. Shangguan et al. (2019) considered the evaluation of trends from reanalysis and
found some problems with the homogeneity of this type of data. Occurrence of the artificial
discontinuities is a great problem in trend studies based on reanalyse data, because their
occurrence artefacts the value of ozone trends and their significance. Only the artificial
discontinuities are problematic, not the real ones. To our best knowledge there are no real
discontinuities in ozone time series of monthly data so the majority of discontinuities is
artificial.
Krizan et al. (2019) searched for the discontinuities in MERRA-2 ozone data. The aim of
our paper is to extend this analysis to the ERA5 and JRA-55 data and compare the results from
all three reanalyses. We use standard Pettitt homogeneity test (Pettitt, 1979) for all reanalyses
in the period 1980–2017. Our study can help us to compare the data quality in these reanalyses
as a first step toward to the trend studies. This paper is divided into the following sections:
Section 2 describes the data and method, Section 3 provides results, Section 4 discusses the
results, and Section 5 contains conclusions.

## 2. Data and method

At first we must explain the Pettitt homogeneity test.
Figure 1 and Table 1 present the artificial time series, which consists of part with no trend
(from member 1 to 5) and part with the negative trends (from member 6 to 10). They model the
onset of Antarctic ozone hole.  Above member 10 we repeat the series and we add 10 to each
member to get a discontinuity. So we have series of the length 20 and with discontinuity. Then
the series is sorted in ascending order. The results are given in the second row of Table 1. Now
we look at the first element in the second row: it is 1, 1 is the $10^{th}$ element of the series (row 1)
and this value is the first element in row 3 of Table 1. Similarly, the second element of the row
2 is 3, it is the $9^{th}$ element of artificial series and this number is the second element of the row
3 in Table 1. When this procedure is done for all elements of the series, we obtain the third row
of Table 1.  Then we perform the cumulative sum of row 3. The results are given in row 4 of
Table 1. Let $X_i$ is the $i^{th}$ element of this cumulative sum. The value $U_i$ is defined as follows:
$$U_i = abs\ (2X_i - i(l+1)) \tag{1}$$

Where abs is the absolute value and l is the length of series (in our case l = 20). The values U
are given in the last row of Table 2. Now we find the maximal value of U from Table 1. It is 81
and it is the $11^{th}$ element of U series.  In next step we must compute the characteristics p is
calculated:
$$P = e^{T} \tag{2}$$

where          $$T = -6(max(U))^2/(l^3+l^2) \tag{3}$$



where l is the length of series. If P< 0.05 the discontinuity is detected at the element of series
where U is maximal, otherwise the discontinuity is not detected according the Pettitt (1979).  In
our case max(U)=81, l=20 so T=-4.69 and P=0.0092. 0.0092<0.05 so the discontinuity is
detected at element 11 of the series. The 11$^{th}$ element is the right element where the
discontinuity occurs (vertical red line in Figure 1 and first row in Table 2). So the Pettitt test is
able to detect the discontinuity, but it does not detect discontinuity where negative trends in
time series starts (element 5 and 16). So this procedure will not wrongly detect the onset of
ozone hole as a discontinuity. We also did some simulations concerning the behaviour of this
test in the cases when the discontinuity occurs at the edge of the series. When the discontinuity
was present at every element up to the 18$^{th}$ one, this test was able to detect it
In this paper, we used the ozone monthly means from MERRA-2, ERA5 and JRA-55
above 500 hPa in the period 1980–2017. Table 2 shows that the top layer in the reanalyses is
not the same for various reanalyses: The top modelling level for JRA 55 is 0.1 hPa and for
MERRA-2 and ERA5 0.01 hPa. However, the top available/published level for all parameters
is 1 hPa (JRA-55 and ERA5) and 0.1 hPa (MERRA-2). The layers 4 hPa and 40 hPa are present
only in MERRA-2 and the data from 125 hPa, 175 hPa and 225 hPa are given only in ERA5
and JRA-55, not in MERRA-2. Reanalyses used in this paper cover the whole satellite era. They
include substantial upgrades and changes to the data assimilation system and input data. We
choose these three reanalyses because they are mostly used in atmospheric community and they
are the newest ones at this time.  In MERRA-2 new constraints are applied to ensure the
conservation of global dry-air mass and to close the balance between surface water fluxes
(precipitation minus evaporation) and changes in total atmospheric water (Takacs et al., 2016).
The modified gravity wave scheme substantially improves the model representation of the quasi
biennial oscillation (Molod et al., 2015; Cog et al., 2016). The assimilation of Microwave Limb
Sounder (MLS) temperature retrievals at high-pressure levels (lower than 5 hPa) should
improve the reanalysis at upper levels. The assimilation of MLS stratospheric ozone profiles
and Ozone Monitoring Instrument (OMI) column ozone since the beginning of the Aura
mission in late 2004 also improve the representation of fine-scale ozone features, especially in
the region around the tropopause (Randles et al., 2016). MERRA and MERRA-2 use the Three
Dimensional Variational (3D VAR) assimilation process. MERRA-2 uses regular latitude–
longitude grids from 1000 to 0.01 hPa (1/2° latitude × 5/8° longitude). The warm bias in the
upper troposphere has gradually been decreasing because the observing system has been
improving. A change in observing systems occurred in July 2006, when the GNSS-RO
refractivities were included into JRA-55. The impact of changes in observing systems on the
JRA-55 time series is reduced compared with JRA-25 but the low-frequency variations are
smaller than those of the SSU datasets, especially in the upper stratosphere (Kobayashi et al.,
2015). Three-dimensional daily mean ozone distributions are implemented in JRA-55 for the
period after 1979. It was produced separately from the JRA-55 data assimilation system using
the T42L68 resolution version of the chemistry climate model (CCM) developed at the
Meteorological Research Institute (MRICCM1, Shibata et al., 2005). According to Harada et
al. (2016) the JRA-55 reanalysis well represent the general features of the QBO and SAO. JRA-
55 has also reduced biases in the lower stratosphere compared with JRA-25 except the northern
polar region. JRA-55 uses the Four Dimensional Variational (4D VAR) assimilation process.
JRA-55 uses regular latitude–longitude grids from 1000 to 0.1 hPa (0.562° latitude x 0.562
longitude). Detail description of all reanalyses (included JRA-55 and MERRA 2) can be found
in Fujiwara et al. (2017) and Gelaro et al. (2017). ERA5 uses 4D-Var data assimilation in
CY41R2 of ECMWF's Integrated Forecast System (IFS), with 137 hybrid sigma/pressure
(model) levels in the vertical, with the top level at 0.01 hPa (0.25° latitude x 0.25 longitude).
Atmospheric data are interpolated to 37 pressure levels up to 1 hPa. From 2000 to 2006, ERA5
has a poor fit to radiosonde temperatures in the stratosphere, with a cold bias in the lower





stratosphere. In addition, a warm bias higher up persists for much of the period from 1979.
More details about ERA5 can be found in CDS (2017).
We did our analysis for each grid point. Spatial (longitudinal and latitudinal) averages are
not used, because during every averaging some information is lost. In each grid and each layer,
we used the time series of the ozone concentration and applied the Pettitt homogeneity test to
look for discontinuity in it. In each grid, the Pettitt test estimates only one main (biggest)
discontinuity, so this procedure is not able to detect multiple discontinuities. This test is widely
used in the climatological research especially for precipitation and temperature analysis (Javari,
2016; Firat et al., 2010; Wijngaard et al., 2003; Kozubek et al, 2020). We are interested
primarily in the spatial distribution of the discontinuities. In each layer and month, we
constructed the map of discontinuity occurrence. The Pettitt test tells us the year in which the
discontinuity in time series occurs. But it does not say how big the discontinuity is or how it
can affect trends. Small discontinuities have little impact on the trend analyses. On the other
hand, a large discontinuity can have a strong trend impact, so we must divide the discontinuities
according to their size. We tried to identify which ones were significant (in our case big enough
to impact the trend) or insignificant according to this rule: Suppose we have a time series with
the length L, and let in year x, the discontinuity occurs. We compute the difference between the
average before the year x and after this year. If this difference is larger than the variance of time
series, we can say that the discontinuity is significant and could have impact on trends, and we
should be careful using this grid point in a trend analysis.
**3.   Results**
For each layer and each reanalysis we compute the average, maximal and minimal DO
from all months and the results are given in Figure 2 (3) which shows the vertical profiles of
the average, minimal and maximal discontinuity occurrence for MERRA-2 (upper panel),
ERA5 (middle panel) and JRA-55 (lower panel) for all (significant) discontinuities. The
percentage shown in Figures 2 and 3 means how many of all grid points at a given level contain
discontinuities, and this is shown for all levels in the form of profile. These profiles enable us
to compare discontinuity occurrence (DO) among the reanalyses. In Figures 2 and 3 there are
layers where the discontinuities are present more frequently or less frequently.
**3.1. Comparison of the discontinuity occurrence between MERRA-2 and ERA5**
**3.1.1. All cases**
The average DO from each month and each common layer is shown in Figure 4 for
MERRA-2 and ERA5 for all discontinuities (upper panel). The sign S in the figures means the
difference between the average DO from MERRA-2 and ERA5 is statistical significant at the
95 % level.  The maximal DO for MERRA-2 occurs at 1 hPa (93,4 %) and minimal at 20 hPa
(24,9 %). The other area of high DO is the troposphere with maximum 85,2 % at 400 hPa.
The ERA5 average DO have also maximum in the upper stratosphere (98,5 % at 2 hPa), sharp
minimum at 5 hPa (25,0 %) and the high DO is observed in the troposphere with maximum
70,4 % at 350 hPa. Above 350 hPa the average DO is higher at the majority of layers for
ERA5. The only statistically significant differences between the average DO is seen at 2, 3
and 250 hPa. On the contrary below 350 hPa we observe higher average DO for MERRA, but
these differences are not significant.
Table 3 presents the average DO difference between MERRA-2 and ERA5 for each
month and each common layer for all discontinuities (left columns). When the differences are



192 positive it means DO is larger in MERRA-2 than in ERA5. The opposite is true for the
193 negative ones. All differences above 1 % in absolute value must be regarded as significant,
194 because the number of grids is very high (1038240 for ERA5 and 207936 for MERRA-2).
195 The DO differences are similar at the same layer and some layers have larger differences than
196 others. At 2 and 3 hPa we see larger and significant negative DO differences. This means the
197 DO is larger in ERA5 than in MERRA-2. The variance of DO is at some layers high, so it is
198 the reason why the differences between MERRA-2 and ERA5 are insignificant at the majority
199 of layers.
200   We can look at the differences in the distribution of discontinuities for areas where the
201 DO differences are significant. We display the results only for month with the highest
202 difference so we must compare the panels within one figure, not among figures. Figure 3
203 (upper panel) reveals two main areas of significant DO differences: upper stratosphere and
204 250 hPa. Figure 5 shows the geographical distribution of discontinuities at 3 hPa for
205 September (difference -67.9 %) for MERRA-2 (upper panel) and for ERA5 (lower panel).
206 The yellow colour means there is no discontinuity in a given grid and the red one shows the
207 discontinuity in a given grid. MERRA-2 displays the majority of grids with no discontinuity,
208 while for ERA5 discontinuities occur for large number of grids. At the 250 hPa level
209 (FigureS1 in supplement) the geographical distribution is similar (difference is smaller -41.5
210 %, April).
212   **3.1.2. Significant cases**
214   Figure 4 shows the average DO from each month and each common layer also for the
215 significant discontinuities (lower panel). The vertical profile of DO has similar shape as in the
216 case of all discontinuities. MERRA-2 has maximal DO at 1 hPa (86.1 %) and minimal at 7
217 hPa (1.5 %). The ERA5 average DO has reached maximum in the upper stratosphere (87.7 %
218 at 2 hPa) and sharp minimum at 5 hPa (8.4 %). The ERA5 average DO in the troposphere is
219 much higher than for MERRA-2 with maximum 48.1 % at 350 hPa. In the case of the
220 significant discontinuities at the majority of layers we observe higher average DO for ERA5
221 than for MERRA-2. The largest differences in the vertical profile pattern between all and the
222 significant discontinuities is seen in the troposphere, where in the case of all discontinuities
223 we see higher DO for MERRA-2 than for ERA5. The opposite is true for the significant DOs.
224 The DO differences between MERRA-2 and ERA5 are significant in the upper stratosphere at
225 3 hPa and in the troposphere at 250, 300 and 350 hPa.
226   Table 3 gives the average DO differences between MERRA-2 and ERA5 for each month
227 and each common layer for the significant discontinuities (right columns). The largest
228 differences are seen at 3 hPa, where MERRA-2 DO is much smaller than that of ERA5. The
229 negative differences are larger in absolute value for the significant discontinuities. In the
230 troposphere we see positive differences in the case of all discontinuities and the negative ones
231 for the significant discontinuities.
232   We can look at the differences in the geographical distribution of discontinuities for the
233 selected cases where DO differences are significant. Figure 3 (lower panel) reveals two main
234 areas of significant DO differences: the upper stratosphere and the troposphere. The
235 geographical distribution of discontinuities at 3 hPa for June (difference -74.8 %) for
236 MERRA-2 and ERA5 is shown in Figure 6. In the case of MERRA-2 the majority of grids
237 reveal no discontinuity, while for ERA5 we observe discontinuities at a large number of grids.
238 At 250 hPa (Figure S2 in supplement) the geographical distribution is similar (difference is
239 smaller -49.1 %, May).





**3.2. Comparison of the discontinuity occurrence between MERRA-2 and JRA-55**

**3.2.1. All cases**

The vertical profiles of DO for MERRA-2 and JRA-55 are seen in Figure 7 (upper panel). Above 10 hPa DO is higher for JRA-55 than for MERRA2 with significant differences at 3 and 5 hPa. Below 10 hPa DO from JRA-55 is smaller than from MERRA2 at all layers. Significant differences occur at 30 hPa and in the troposphere below 300 hPa, where difference in each layer is significant. These results are supported also by Table S1 in supplement, where the average monthly DOs are shown. These differences are negative above 10 hPa and positive below with maximal differences in the stratosphere at about 5 hPa and in the troposphere. For the majority of grids there are no discontinuities in the case of MERRA-2. The opposite is true for JRA-55. In the troposphere the situation is very different. Figure S3 in supplement displays the geographical distribution of DO at 400 hPa in March (difference 64.8 %). There is a discontinuity in the case of MERRA-2 (upper panel) at nearly all grids. DO is substantially lower for JRA-55 (lower panel).

**3.2.2. Significant cases**

In the case of the significant discontinuities (Figure 7, lower panel) we can observe at the majority of layers JRA-55 DO to be smaller than that of MERRA2. There are huge differences in behaviour of DO in the case of significant and all discontinuities in the uppermost layers (1 and 2 hPa). DO of all discontinuities is larger for JRA-55, the opposite is true for the significant discontinuities, where at 1 hPa the MERRA-2 versus JRA-55 differences are significant. From 3 hPa down to 10 hPa we observe insignificant differences with higher DO values for JRA-55. Below 10 hPa DO is higher for MERRA-2 at all layers, but these differences are insignificant except for 50 hPa. The monthly DO values are shown in Table S1 in supplement. At 1 hPa DO differences for all discontinuities are small and negative for nearly all months, while for the significant discontinuities these values are high and positive (DO is higher for MERRA-2). Similar DO patterns are seen at 2 hPa. From 3 hPa down to 10 hPa we observe small negative monthly DO values in agreement with Figure 7. Below 10 hPa DO values are positive for majority of months and layers. Figure 8 shows the geographical distribution of significant discontinuities at 1 hPa for October (difference 84.9 %). In MERRA-2 there are discontinuities at the vast majority of grids, while for JRA-55 the occurrence of discontinuities is strongly reduced.

**3.3. Comparison of discontinuity occurrence between ERA5 and JRA-55**

**3.3.1. All cases**

Figure 9 (upper panel) shows vertical profiles of the average DO for ERA5 and JRA-55 for all discontinuities. The profile patterns are similar as for MERRA-2 and JRA-55 DOs. Above 5 hPa DO for ERA5 is comparable or slightly higher than for JRA-55. At 5 hPa there is a sharp minimum in DO in the case of ERA5, so this difference is significant. Below 7 hPa DO for JRA-55 is smaller at all layers. These differences are significant from the lower stratosphere (below 225 hPa) down to the upper troposphere (above 500 hPa). These results are confirmed also by Table S2 in supplement, where monthly values of DO are shown. Above 5 hPa we observe small negative differences (DO in JRA-55 is higher). At 5 hPa these differences are the highest in absolute value. Below 7 hPa the differences are positive at the





majority of layers and months. The geographical distribution of all discontinuities at 5 hPa for
August (difference -83.3 %) reveals Figure 10 for ERA5 (upper panel) and for JRA-55 (lower
panel). For JRA-55 the discontinuities occur at vast majority of grids, while for ERA5 the
number of discontinuities is substantially lower. The situation is opposite in the troposphere
(Figure S4 in supplement). At 250 hPa in June (difference 58.7 %) the occurrence of
discontinuities is higher for ERA5 than JRA-55.

### 3.3.2. Significant cases

Figure 9 (lower panel) shows the average vertical profile of DOs for ERA5 and JRA-55
for the significant discontinuities. Again the vertical distribution of DOs is similar to that for
MERRA-2. At all layers except 5 hPa the discontinuity occurrence is higher for ERA5 than
JRA-55. These differences are significant above 3 hPa, at 50 and 70 hPa and at all layers
below 225 hPa. These conclusions are in agreement with the results of Table S2 in
supplement, where at the majority of months and layers we see positive differences, which
means DO is higher for ERA5 than JRA-55. Figure 11 (2 hPa, June difference 91.6 %) and
Figure S5 in supplement (400 hPa, December, difference 54.7 %) also support these
conclusions.

### 4. Discussion

The vertical profile of DO is similar at each month within one reanalysis, which means it
is reasonable to compare the DO profiles among the reanalyses. There are layers at which DO
is higher (lower) than at the others and these layers are the same for each month within one
reanalysis. In general, DO is higher in the upper stratosphere and the other area of the high
DO is in the troposphere.  The errors of satellite measurements increase in the lowest
stratosphere and in the troposphere, where the most important measurements are those from
the sondes, but they are pointwise. When the amount of data is lower, the changes in data
amount or changes in measurement techniques have greater impact on the reanalysis result
and on discontinuity occurrence, which might be the reason for the tropospheric increase of
DO.
Now we discuss the results of comparing the discontinuity occurrence for each pair of
reanalyses. When we look at the DO in the uppermost layers we see DO for MERRA-2 is
lower than those of ERA5 and JRA-55 for all discontinuities. Discontinuity occurrence is very
high at about 1 hPa at all three reanalyses. MERRA-2 has also lower DO for significant
discontinuities than ERA5 in these layers. It is very interesting DO for the significant cases is
lower for JRA-55 than that for MERRA-2 and ERA5 above 5 hPa.
MERRA-2 and ERA5 have the same top model layer (0.01 hPa), but for ERA5 the top
model layer available for public is 1 hPa. ERA5 and JRA-55 use the same assimilation
procedure (4D VAR), while MERRA-2 uses 3D VAR procedure. The differences in upper
stratosphere, which means around top of all reanalyses, are one of the main problem which
should be studied in more details. This region is very important for vertical coupling in the
middle atmosphere. DO differences among the reanalyses could be affected by differences in
data used in these reanalyses.  On the other hand, amount of observations in this region is very
low compared to lower pressure levels, which contributes to large DO in the upper
stratosphere. That is why assimilation procedure and used models can play important role.
Due to combination of different assimilation procedure, not ideal satellite and homogenous
observations and complicated modelling processes it is very difficult to identify the main
problem.





Another area of DO differences is the troposphere. Occurrence of all discontinuities is
higher for MERRA-2 than for ERA5, but for the significant discontinuities, DO is much
higher for ERA5 than for MERRA-2. It means there are more insignificant discontinuities in
MERRA-2 than in ERA5, but the opposite is true for the significant ones. For trend analyses
the significant discontinuities have larger impact on result, so the ERA5 data in the
troposphere is more suitable for trend analysis than that of MERRA-2. DO values from JRA-
55 are the smallest from all three reanalyses at each layer below 10hPa for all and the
significant discontinuities. So the JRA-55 is the most suitable for ozone trend analyses from
all three analyses due to small number of significant discontinuities, which affect the trend
results.
It is necessary to focus on the connection between the results of this paper and future
trend analyses. The greatest impact on trend results has got the presence of the significant
discontinuities in the time series. According to our results the number of these significant
discontinuities is the lowest in the JRA-55. This number is higher in MERRA-2 and ERA5
and it is comparable between these two reanalyses. If one wants to explore reanalysis data for
trend analyses, it is necessary to look at the correlation between the trend patterns and patterns
of discontinuity occurrence. If the correlation is present, the discontinuity influence must be
taken into account. It will be interesting to look at the discontinuity occurrence in the total
ozone time series from reanalyses, because we did this analyses for the ozone concentration
time series from each layer, not for total ozone. It is reasonable to suppose the results might
be partly different.
Wargan et al. (2018) used three different MERRA-2 assimilation and model products and
found downward lower stratospheric ozone trends from 1998 to 2016 similar to satellite-based
observational trends by Ball et al. (2018).
A significant change to the MERRA-2 meteorological assimilation occurred in 1998 with
the launch of ATOVS (Long et al., 2017) onboard NOAA-15. The changes associated with
new ATOVS data are also evident in other reanalysis systems (e.g., ERA-Interim and JRA-
55; Long et al., 2017). 1998-1999 has been identified as the period of greatest change to
reanalysis meteorology in the middle atmosphere (Fujiwara et al., 2017).
In our paper we were not interested in temporal occurrence of discontinuities. According
to Shangguan et al. (2019) the first period of ozone discontinuity presence is about 2003 when
MERRA-2 switched from SBUV to MLS and the other in 2015 when MERRA-2 and ERA5
started to use 4.2 MLS data instead of version 2.2. Similar result for MERRA-2 was obtained
by Krizan et al. (2019).

**5.  Conclusions**
The occurrence of discontinuities in series of ozone concentration data at various
stratospheric and tropospheric levels between 500 and 1 hPa were searched for three modern
reanalyses ERA5, MERRA-2 and JRA-55. This study is based on analysis of data in
individual grid points, not on zonal or other averages. The obtained results have implications
for usability of ozone data from reanalyses for trend studies. The main results of this paper are
as follows:
▪  There are differences in the discontinuity occurrence frequency among reanalyses.
▪  In the upper stratosphere we observed the tendency toward the higher discontinuity
occurrence in all reanalyses

▪  Another area of higher discontinuity occurrence is the troposphere



- ▪ The discontinuity occurrence in JRA-55 is on average the lowest from all three reanalyses below 10 hPa, especially for the significant discontinuities.

- ▪ According to our results, JRA-55 is the most suitable for reanalyse trend studies due to the low significant discontinuity occurrence.

The follow-on investigations should focus on a similar investigation of discontinuity occurrence in the total ozone from reanalyses and on application of reanalyses data for long-term trend studies.

**Acknowledgements**

ECMWF, NASA and Japanese Meteorological Society are acknowledged for development of ERA5, MERRA-2 and JRA-55 and for possibility to use these data.

**Author Contributions:** This paper was prepared in close collaboration of all authors but the major part of the work was done by Peter Krizan.

**Competing interests:** The authors declare no conflicts of interest.

**Financial support:** Support by the Czech Science Foundation via Grant 18-01625S is acknowledged.

**Data availability:**

**ERA5:** https://cds.climate.copernicus.eu/cdsapp#!/home

**JRA-55:** https://rda.ucar.edu/datasets/ds628.0/#access

**MERRA2:** https://cmr.earthdata.nasa.gov/search/concepts/C1276812931-GES_DISC.html

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

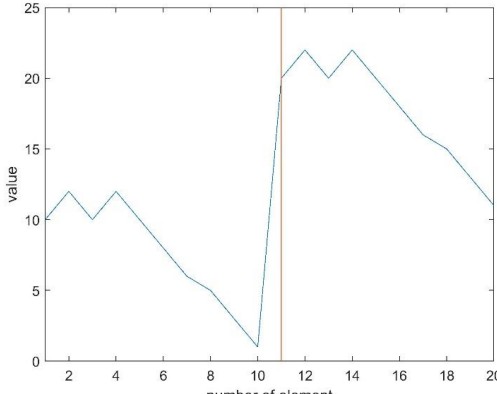

**Figure 1** Artificial time series with discontinuity at element 11. The vertical red line is the
resulting discontinuity searched for by Pettitt homogeneity test.



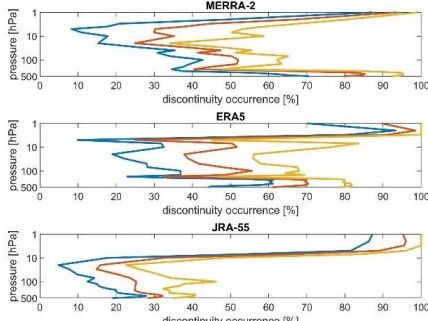



**Figure 2** Vertical profiles of the minimal (blue), average (red) and maximal (orange)
discontinuity occurrence for MERRA-2 (upper panel), ERA5 (middle panel) and JRA-55
(lower panel) for all discontinuities.


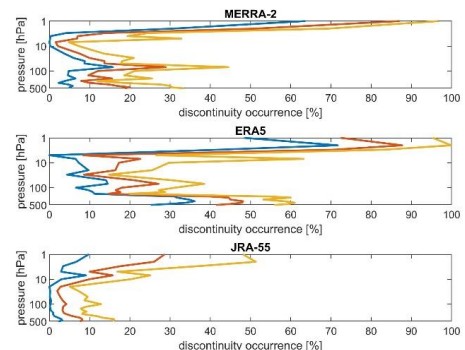


**Figure 3** The same as Figure 2 but for the significant discontinuities.


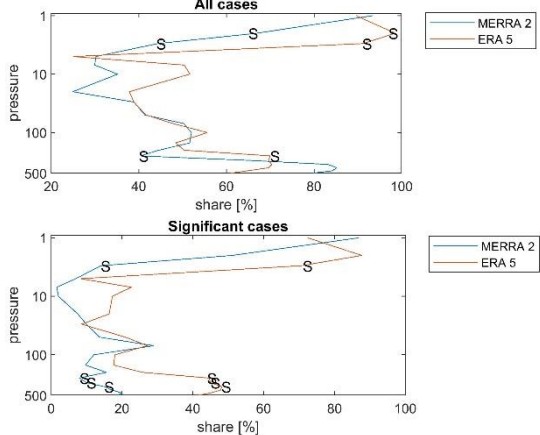


**Figure 4** The average vertical profile of DO for MERRA-2 and ERA5 for all (upper panel) and significant discontinuities (lower panel).  Letter S means significant difference between ERA5 and MERRA-2.






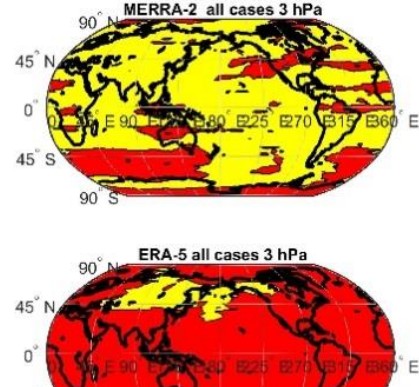


**Figure 5** Geographical distribution of all discontinuities (yellow – no discontinuity, red – discontinuity) for MERRA-2 (upper panel) and ERA5 (lower panel) at 3hPa for September.



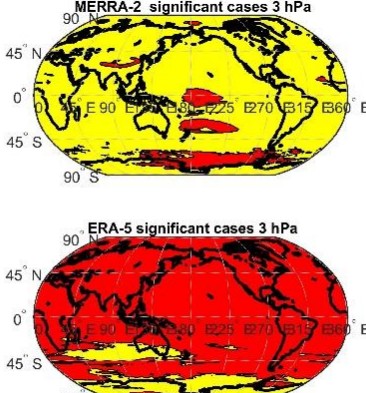

**Figure 6** Geographical distribution of   the significant discontinuities for MERRA-2 (upper panel) and ERA5 (lower panel) at 3hPa for June.







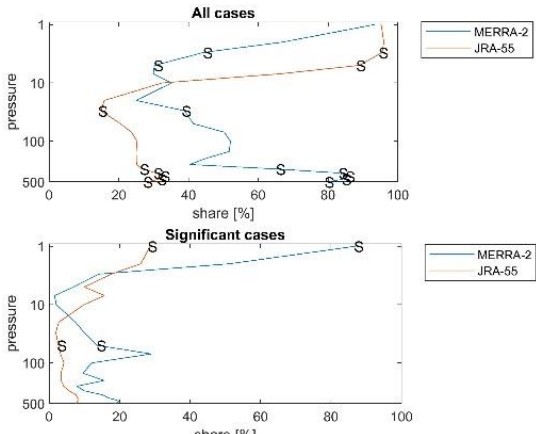

**Figure 7** The average vertical profile of DO for MERRA-2 and JRA-55 for all (upper panel) and significant discontinuities (lower panel).




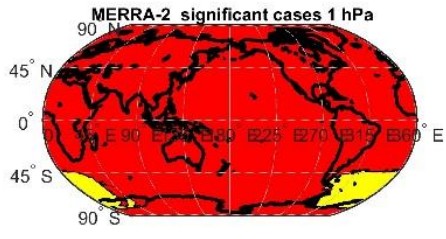

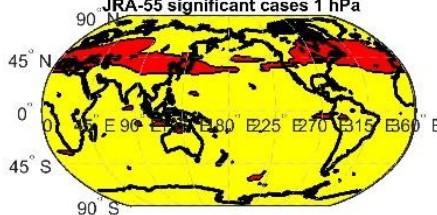

**Figure 8** Geographical distribution of the significant discontinuities for MERRA-2 (upper panel) and JRA-55 (lower panel) at 1hPa for October.







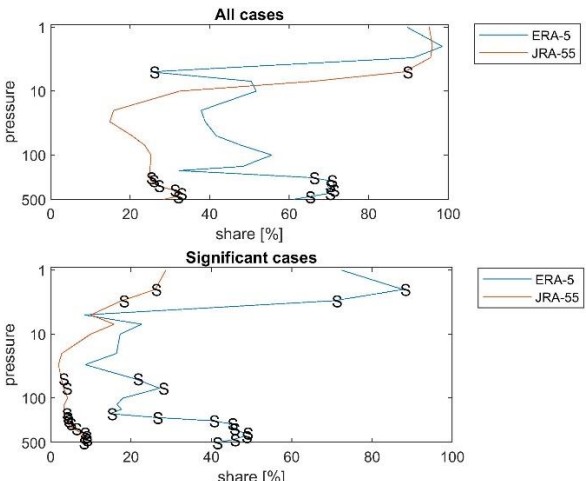



**Figure 9** The average vertical profile of DO for ERA5 and JRA-55 for all (upper panel) and
significant discontinuities (lower panel).



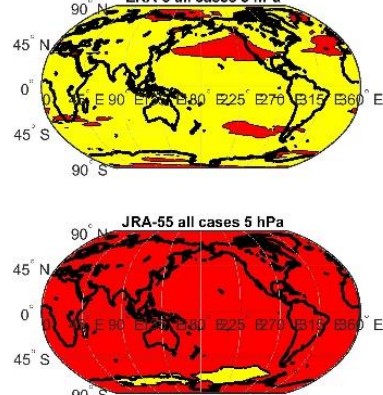


**Figure 10** Geographical distribution of all discontinuities for ERA5 (upper panel) and JRA-55
(lower panel) at 5hPa for August.






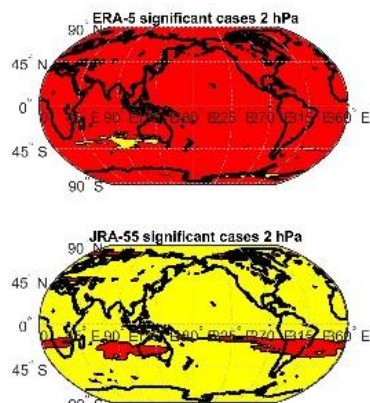

**Figure 11** Geographical distribution of the significant discontinuities for ERA5 (upper panel)
and JRA-55 (lower panel) at 2hPa for June.

**Table 1** Values which are needed for explanation of the Pettitt homogeneity test.

| 10 | 12 | 10 | 12 | 10 | 8 | 6 | 5 | 3 | 1 | 20 | 22 | 20 | 22 | 20 | 18 | 16 | 15 | 13 | 11 |
|---|---|---|---|---|---|---|---|---|---|---|---|---|---|---|---|---|---|---|---|
| 1 | 3 | 5 | 6 | 8 | 10 | 10 | 10 | 11 | 12 | 12 | 13 | 15 | 16 | 18 | 20 | 20 | 20 | 22 | 22 |
| 10 | 9 | 8 | 7 | 6 | 1 | 3 | 5 | 20 | 2 | 4 | 19 | 18 | 17 | 16 | 11 | 13 | 15 | 12 | 14 |
| 10 | 19 | 27 | 34 | 40 | 41 | 44 | 49 | 69 | 71 | 75 | 94 | 112 | 129 | 145 | 156 | 169 | 184 | 196 | 210 |
| 1 | 4 | 9 | 16 | 25 | 44 | 59 | 70 | 51 | 68 | 81 | 64 | 49 | 36 | 25 | 24 | 19 | 10 | 7 | 0 |


**Table 2** Layers in reanalyses used in this paper.

| hPa | 0,1 | 0,3 | 0,4 | 0,5 | 0,7 | 1 | 2 | 3 | 4 | 5 | 7 | 10 | 20 | 30 | 40 | 50 |
|---|---|---|---|---|---|---|---|---|---|---|---|---|---|---|---|---|
| MERRA -2 | * | * | * | * | * | * | * | * | * | * | * | * | * | * | * | * |
| ERA -5 | | | | | | * | * | * | | * | * | * | * | * | | * |
| JRA -55 | | | | | | * | * | * | | * | * | * | * | * | | * |


| hPa | 70 | 100 | 125 | 150 | 175 | 200 | 225 | 250 | 300 | 350 | 400 | 450 | 500 |
|---|---|---|---|---|---|---|---|---|---|---|---|---|---|
| MERRA -2 | * | * | | * | | * | | * | * | * | * | * | * |
| ERA -5 | * | * | * | * | * | * | * | * | * | * | * | * | * |
| JRA -55 | * | * | * | * | * | * | * | * | | * | * | * | * |










**Table 3.** The differences in the discontinuity occurrence between MERRA-2 and ERA5 in
individual months for all (left column) and significant (right column) discontinuities. The
positive difference means higher number of discontinuities in MERRA-2 than in ERA5.

| hPa | January All | January Sig. | February All | February Sig. | March All | March Sig. | April All | April Sig. | May All | May Sig. | June All | June Sig. |
|---|---|---|---|---|---|---|---|---|---|---|---|---|
| 1 | -2.3 | 8.5 | 28.5 | 39.4 | 11.3 | 25.5 | 10.5 | 7.8 | -9.9 | -7.2 | 3.2 | -13.2 |
| 2 | -25.2 | -25.2 | -51.2 | -55.3 | -39.8 | -40.8 | -28.1 | -20.5 | -19.6 | -43.7 | -29 | -67.7 |
| 3 | -37.4 | -59.2 | -47.1 | -54.7 | -65.0 | -60.6 | -40.9 | -58.7 | -39.0 | -63.4 | -38.6 | -74.8 |
| 5 | 8.4 | -11.0 | 18.9 | 24.4 | 22.9 | 0.2 | -8.8 | -9.7 | -15.0 | -17.1 | -6.3 | -0.4 |
| 7 | 14.6 | -6.5 | 12.9 | -16.7 | -10.0 | -19.7 | -33.1 | -10.2 | -53.2 | -37.4 | -56.1 | -62.7 |
| 10 | 1.1 | -22.4 | 7.6 | -20.7 | 9.0 | -18.5 | -27.4 | -15.5 | -31.4 | -17.7 | -29.5 | -16.9 |
| 20 | 10.3 | -1.6 | -0.9 | -9.3 | -13.8 | -1.1 | -5.8 | -5.7 | -22.4 | -11.8 | -30.7 | -20.1 |
| 30 | 4.0 | 0.5 | 7.2 | 0.2 | 2.3 | 2.5 | -8.0 | -3.0 | -11.4 | -5.6 | -15.0 | -2.6 |
| 50 | -8.2 | -5.1 | 5.4 | 0.0 | 4.3 | -6.4 | 1.4 | -11.4 | 12.5 | -0.6 | 5.1 | -12.0 |
| 70 | 5.7 | 0.9 | 13.2 | 1.8 | 5.0 | 5.1 | 8.8 | 7.9 | 10.8 | 17.5 | 7.1 | 14.9 |
| 100 | -15.7 | -7.3 | -19.7 | -9.7 | -2.3 | -2.8 | 5.9 | -1.8 | -8.0 | -2.2 | 5.3 | 2.7 |
| 150 | 14.8 | 0.3 | 8.1 | -5.1 | -4.9 | -7.1 | 1.7 | -6.5 | -11.6 | -9.5 | -4.7 | -10.1 |
| 200 | -4.5 | -6.9 | -0.1 | -0.1 | -11.6 | -12.5 | -13.8 | -9.1 | -17.3 | -17.5 | -9.1 | -15.5 |
| 250 | -22.2 | -36.9 | -24.6 | -32.1 | -39.1 | -44.8 | -41.5 | -43.8 | -40.2 | -49.1 | -40.2 | -42.1 |
| 300 | 4.4 | -30.1 | 7.9 | -25.3 | -9.0 | -38.8 | -6.8 | -37.6 | -12.7 | -38.9 | -32.9 | -44.6 |
| 350 | 9.0 | -29.0 | 22.7 | -19.4 | 28.5 | -29.1 | 30.9 | -29.2 | 12.1 | -38.1 | -11.6 | -44.3 |
| 400 | 14.5 | -25.8 | 27.4 | -16.5 | 35.7 | -21.4 | 37.7 | -28.1 | 12.0 | -33.0 | -7.1 | -40.7 |
| 450 | 12.2 | -21.5 | 18.4 | -26.2 | 40.7 | -15.9 | 38.9 | -25.1 | 24.5 | -14.2 | 5.3 | -30.4 |
| 500 | 3.2 | -30.7 | 11.3 | -24.0 | 31.8 | -11.6 | 29.9 | -21.9 | 25.5 | -7.2 | 23.7 | -17.8 |



















Table3 continuation

| hPa | July All | July Sig. | August All | August Sig. | September All | September Sig. | October All | October Sig. | November All | November Sig. | December All | December Sig. |
|---|---|---|---|---|---|---|---|---|---|---|---|---|
| 1 | 2.8 | 18.9 | -5.9 | 11.8 | 3.2 | 8.5 | 3.3 | 16.3 | 0.9 | 28.5 | 0.0 | 29.5 |
| 2 | -39.7 | -29.1 | -26.3 | -32.5 | -22.7 | -37.4 | -17.8 | -18.3 | -34.8 | -26.2 | -45.5 | -34.0 |
| 3 | -47.6 | -63.1 | -63.4 | -63.3 | -67.9 | -66.0 | -50.0 | -42.0 | -34.6 | -22.9 | -30.4 | -49.6 |
| 5 | 14.7 | 5.2 | 10.1 | -0.1 | -4.3 | 0.6 | -2.3 | -6.6 | 3.5 | 0.6 | 20.7 | -3.5 |
| 7 | -46.0 | -39.5 | -32.4 | -28.0 | -24.7 | -11.2 | -20.2 | -6.0 | -7.9 | -9.9 | 10.6 | -5.1 |
| 10 | -37.7 | -17.4 | -29.3 | -13.3 | -16.4 | -9.2 | -21.7 | -6.5 | -11.3 | -5.1 | -10.9 | -19.5 |
| 20 | -35.3 | -18.1 | -30.1 | -13.7 | -8.7 | -5.5 | -6.0 | -7.2 | -10.0 | -9.7 | -1.3 | -2.9 |
| 30 | -14.1 | -6.4 | 10.5 | 4.5 | 6.3 | 13.0 | 13.2 | 4.2 | 13.4 | 5.6 | -7.9 | 3.2 |
| 50 | 0.4 | -10.2 | -4.9 | -19.5 | 11.6 | 0.1 | 3.6 | 0.4 | -15.1 | -10.0 | -19.4 | -13.2 |
| 70 | -3.2 | 6.0 | -6.6 | -6.1 | -5.7 | -7.0 | 1.5 | 0.7 | 0.6 | -7.8 | -7.9 | -9.0 |
| 100 | -1.9 | -1.3 | 0.6 | -21.0 | 2.1 | -11.0 | -3.8 | -2.5 | -1.2 | -5.0 | -4.9 | -9.3 |
| 150 | 3.4 | -3.5 | -10.2 | -23.3 | -2.1 | -18.0 | 16.4 | -1.6 | 8.4 | -9.4 | 19.0 | -1.7 |
| 200 | 0.7 | -1.3 | 5.3 | -7.5 | 5.0 | -16.8 | -3.8 | -12.0 | -13.5 | -12.9 | -7.1 | -10.8 |
| 250 | -37.3 | -34.1 | -21.7 | -29.5 | -20.0 | -27.7 | -14.8 | -21.4 | -30.0 | -42.9 | -23.8 | -35.6 |
| 300 | -42.0 | -47.3 | -0.2 | -34.5 | 2.8 | -29.9 | 14.8 | -25.8 | 17.3 | -38.1 | 9.9 | -29.3 |
| 350 | -25.9 | -53.6 | 14.6 | -37.4 | 13.8 | -34.5 | 22.0 | -26.7 | 17.4 | -31.0 | 17.9 | -25.6 |
| 400 | -10.5 | -40.5 | 11.7 | -36.4 | 15.3 | -36.6 | 22.9 | -29.9 | 13.6 | -34.0 | 13.8 | -27.6 |
| 450 | 14.7 | -17.6 | 15.4 | -23.9 | 13.4 | -34.9 | 20.5 | -29.7 | 13.1 | -34.8 | 14.9 | -25.9 |
| 500 | 24.7 | -1.7 | 20.6 | -16.4 | 10.6 | -36.6 | 16.2 | -28.6 | 9.9 | -38.9 | 16.4 | -34.6 |
