# Peer review of "Occurrence of discontinuities in the ozone concentration data from three reanalyses"

_Atmospheric Chemistry and Physics, 2020_

## Referee Comment (RC1) · Anonymous Referee #1 · 6 Sep 2020

**Review of "*Occurrence of discontinuities in the ozone concentration data from three reanalyses*" by P. Krizan, M. Kozubek, and Jan Lastovicka**

This manuscript aims to identify discontinuities in the ozone records from three modern reanalyses, MERRA-2, ERA-5, and JRA-55 to inform future ozone trend studies. Discontinuity detection is done by calendar month using Pettitt's test applied separately to monthly ozone time series at each reanalysis grid point between 500 hPa and 1 hPa. The paper shows maps of spatial discontinuity occurrences (DO) and global DO profiles from each reanalysis. It also compares the DO counts as well as significant (as defined in the paper) DO counts among the reanalyses. All discontinuities are assumed to be spurious and, therefore, deleterious to trend studies. Based on these results, the study finds large differences in DO counts among the reanalyses but, generally, with more DO in the upper troposphere and upper stratosphere than between these layers. JRA-55 ozone shows the least number of DOs and is deemed most suitable for trend studies.

A comprehensive quantitative assessment of spurious ozone variability in major reanalyses would be extremely useful and timely, given a great deal of scientific interest in trends. Some initial evaluation of reanalyses' ozone (as well as temperature) discontinuities has been done as part of the SPARC Reanalysis Intercomparison Project but more systematic approach is needed. It's really nice to see that the Authors of this manuscript recognize the importance of this kind of work and have taken a step toward addressing the problem quantitatively. Unfortunately, I don't find this study in its current form suitable for publication. The main issue is that I'm not convinced that the DO detection algorithm actually works as intended or that it produces useful results in the context of trend studies. In particular, DO detection appears to miss some very significant step changes present in MERRA-2 ozone. I provide several examples below. Furthermore, I find the description and discussion of the methods used insufficient. There is no discussion of the assumptions of Pettitt's test and, more generally, applicability of this test to ozone data with their substantial interannual variability and time-dependent trends. A simple statement that this method has been used before in different contexts is not sufficient. No justification is given for the Authors' definition of '*significant discontinuity*'. I'm not convinced that that definition is useful given the small magnitudes of ozone recovery trends that we want to estimate. Furthermore, the paper assumes that all DOs are spurious but fails to offer any justification of this assumption that I find problematic – or at least non-obvious. Finally, I cannot agree with the concluding statement about JRA-55's superior suitability for ozone trend studies. JRA-55 does not assimilate ozone profile data, which makes it fundamentally different than the other two reanalyses. I explain all these issues in detail in my general comments.

I'm afraid I cannot recommend this manuscript for publication in ACP; not without some very significant rewriting. I imagine that would require a serious rethinking of the methodology, a solid justification of all the assumptions made, and making a connection to both geophysical processes affecting ozone and the observations that the reanalyses use. It will be a very different paper, but I think it could be a very important one. I offer some suggestions in my general comments below.

Finally, there's always a chance that I misunderstood something fundamental and some of my criticism is unwarranted. I'm looking forward to the Authors' responses. I'll be happy to be proven wrong!

**General comments**

1. I have worked through a few examples that make me think that the discontinuity detection method used in this paper is problematic. All my plots use monthly MERRA-2 ozone data. First, let's look at supplementary Figure S2. It shows a map of the 250-hPa MERRA-2 and ERA-5 May discontinuities that are identified as significant. With the exception of a few isolated patches in the tropics and around 60◦S **it shows no significant discontinuities in MERRA-2 at that level, thus suggesting that the reanalysis ozone at 250 hPa is by and large suitable for trend analyses**. But that is demonstrably incorrect! The plot below shows the 250-hPa MERRA-2 ozone averaged between 60◦S and 60◦N in all seasons (black) and for May only (magenta). Two large step changes are evident even without removing the annual cycle: one in late 2004 (transition from SBUV to EOS Aura) and one in early 2015 (transition from MLS v2.2 to v4.2). There's no doubt that these discontinuities are large enough to affect trends, including global trends.

[Figure]

**Fig R1**

Pursuing this a little further, Supplementary Figure S1 shows all 250-hPa DOs (significant and otherwise) for April. There appears to be no DOs along 45◦S between ~60◦W and 140◦E as indicated by the yellow shading there. Below, I plot the 40◦W–120◦E April average ozone at 45◦S (magenta) along with the time series at a single point at 0◦E (black)

[Figure]

**Fig R2**

Both the single point time series and the average exhibit two sizable jumps, consistent with the ozone observing system transitions, but evidently not detected by the algorithm.

Moving on to the middle stratosphere, Figure 7 indicates that MERRA-2 ozone has almost no significant (i.e. trend-affecting) DOs at 10 hPa, implying that trends can be safely derived at that level. But that, again, is doubtful upon closer examination, although the situation is more subtle here than at 250 hPa. Plotted below is the time series of globally averaged 10-hPa MERRA-2 monthly ozone (black) between 2000 and 2019.

[Figure]

**Fig. R3**

There is a drop in ozone mixing ratio in late 2004 but there is also a lot of variability and, based on this time series alone, it is not clear whether that particular drop is spurious or real. The algorithm used in the paper again does not seem to register many significant discontinuities at 10 hPa. But we have other data sets to compare this one with! The green line shows ozone from MERRA-2's "cousin", GEOS-RPIT/FPIT, a reanalysis-like product that is very similar to MERRA-2 except that it continues to use SBUV ozone past 2004, while MERRA-2 switches to MLS and OMI data. Alternatively, we could use JRA-55 or even a CTM simulation for this purpose just as well; any data set that does not have an

observing system change in 2004 would do. It's clear that while the MERRA-2 and RPIT lines in the figure above are initially very close, they sharply diverge in 2004. Since there was no observing system change in RPIT in 2004, the difference most likely result from MERRA-2 switching from SBUV to Aura data and represents a spurious discontinuity. Again, this discontinuity is large enough to affect trend calculations (if not removed). And again, it is overlooked by the algorithm used in this study. I found similar examples at 3 hPa.

Another example. According to Fig. 7 JRA-55 has only a tiny fraction of significant DOs between 200 hPa and 20 hPa. Compare that with Fig. 11 of Davis et al. 2017. The right-hand panel shows the differences between 70-hPa reanalysis ozone and a homogenized satellite-derived product. JRA-55 exhibits step changes of between 15% and ~40% in the tropics and NH (lower stratosphere is where we would expect impacts from assimilation of total ozone observations). The largest discontinuity that coincides with the TOMS-OMI transition affects more than half the globe and is an order of magnitude larger than lower stratospheric trends per decade! That seems inconsistent with Fig. 7. Those two results have to be reconciled.

To learn more about the DO detection method I also looked at the predecessor paper, Krizan et al. 2019, that uses the same algorithm to identify MERRA-2 ozone discontinuities (it is the same algorithm, right?), and, unlike the present manuscript, shows temporal distribution of the DOs it detected. Figure 5 of Krizan et al 2019 shows the timings of MERRA-2 ozone discontinuities at 0.1 hPa(*) indicating that the dominant ones occur **in the 1990s**. But the plot of the 60°S–60°N 0.1-hPa ozone average shown below (or we could just pick any individual grid-point location) clearly shows that the dominant (if not the only) discontinuity occurs in 2004 in all months. Nothing remarkable happens in the 1990s - and that makes sense given that ozone data in the mesosphere were suppressed in MERRA-2 prior to assimilation of MLS that started in 2004 (Wargan et al., 2017).

[Figure]

(*) In passing, I note that MERRA-2 ozone in the mesosphere is not suitable for science regardless of any discontinuities; it's basically a simple $O_x$ partitioning scheme (all atomic oxygen in day light, ozone at night) with no observations assimilated prior to 2004. Still, one can apply the algorithm

Why did the algorithm detect nonexistent DOs in the 1990s? Something is wrong. While I'm not reviewing Krizan et al. 2019 here, the present paper does use the same methodology and that methodology clearly produces questionable results. In summary, the algorithm used in this study appears to miss a lot of very large DOs in the reanalysis ozone time series, related to observing system changes, and possibly falsely identifies some DOs that don't really exist as in the last example, making it difficult to trust any of the results showed in the figures and tables. This is a major problem for a study that aims at identifying discontinuities to inform trend analyses. Admittedly, I find some of these findings extremely puzzling (e.g. the failure to detect the rather obvious DOs in Fig. R2) and I'm open to the possibility that I'm doing something wrong. Even though it's hard to mess up straightforward plotting, if anyone is capable of it, I certainly am;). I'm very much looking forward to reading the Authors' response.

2. **Methods**. While intuitive, the notion of 'discontinuity' is not defined in the paper; we are dealing with discrete time series so it's not obvious what is meant by it. I take it to mean a 'change point', $t_c$, such that the ozone pdf for $t<t_c$ is different than for $t>t_c$. I think that is in line with Pettitt's method. If that's true then what happens if the pdf undergoes constant changes, not just at isolated points? Time series of stratospheric ozone exhibit time-dependent trends (rapid decline in the second half of the 20th century, followed by slow recovery, at least at some altitudes/latitudes), and generally can't be modeled as piecewise stationary processes. Does that pose a problem for Pettitt's algorithm? Also, how do those trends affect the classification of discontinuities as 'significant' or not significant? Furthermore, how large does a discontinuity have to be, compared to interannual variability, for the test to detect it? The example provided in Fig 1 features a very large step change. In contrast, the observed and projected ozone recovery trends are of the order of no more that ~1-3% per decade (middle to upper stratosphere), i.e. 0.1–0.3% per year (see e.g., the LOTUS report, SPARC/IO3C/GAW 2019), far less than typical interannual variability. Consequently, even a small DO masked by year-to-year variability can affect trends. Related to that, the significance criterion adopted here (a DO is significant if the difference between the averages before and after the change point exceeds the time series standard deviation; the text says "variance", I take it be a mistake) lacks any justification and doesn't appear to yield meaningful results as demonstrated above. At the very least it should be shown what impacts on trends can be expected from some larger "non-significant" discontinuities vs significant ones. Furthermore, comparing the 'before' and 'after' averages may be deceptive if there is a second discontinuity in the opposite direction (Figs R1 & R2). Again, one should keep in mind that the trends we are trying to estimate are very small.

3. I think that part of the problem is that the approach adopted here, while interesting, is overly simplistic. An algorithm applies a simple statistical test to reanalyses' output but doesn't take advantage of any prior information that we have at our disposal, and that potentially could make DO detection far more accurate:
   - As most reanalyses use "frozen" GCMs and data assimilation schemes any discontinuities arise from changes in their observing systems and, potentially in the boundary conditions, all of which are documented. This is valuable prior information because it tells us where to look for DOs.

- Interannual ozone variability is controlled by a number of observable factors: solar cycle, QBO, ENSO, SSWs, volcanic aerosols, tropopause height. There are modes of variability that operate on a range of time scales and constitute confounding factors in trend and discontinuity detection. But since we know them, we can remove them from ozone time series e.g. using multiple linear regression.
- There's an advantage to using multiple reanalyses, perhaps along with satellite observations and model simulations as data sets can serve as transfer standard for other data sets. For example, to test whether a particular change in the observing system led to a discontinuity in one reanalysis, one could compare that reanalysis with another data set that didn't have that change (e.g. Fig. R3 above).

Consequently, a useful study of spurious jumps in reanalyses ozone could be an extension of the method used here but employ a few extra steps. As a first step, known modes of variability could be removed from ozone time series as it's typically done in trend estimates. Then DOs could be identified in the in the residual signal (much smaller variance) and analyzed in the context of the changes in input observations, potentially using other data sets for comparison. Ideally, the end result would be a spatially resolved catalogue of spurious discontinuities, along with their timings and estimated magnitudes. Researchers could use that information either to avoid deriving trends over certain time periods and regions or, better yet, include it as a prior in trend estimation. There are also ways to remove known discontinuities from time series. I think that would be far more useful than just having maps of DOs (Figs 5, 6, 8, 10, 11) that can only tell us at which grid points not to do trends, especially that if these maps reflected the DOs correctly, they would likely be mostly red.

4. The paper assumes that all DOs are spurious: LL51-53 "*To our best knowledge there are no real discontinuities in ozone time series of monthly data, so the majority of discontinuities is artificial.*" First (and that's me being nit-picky), if there are **no** real discontinuities then **all** of them, not *majority* of them, are artificial. More importantly, how valid is that assumption? The plot below shows March NH polar ozone at 70 hPa from MERRA-2. There are anomalously low values in the mid 1990s. That period featured high sulfate aerosol loading from Mt Pinatubo and, remarkably, no sudden stratospheric warmings between 1990 and 1997. Both factors would act to decrease polar springtime ozone. A cursory look at the figure may suggest a discontinuity in the time series. But if it is a discontinuity, it's likely a real one!

[Figure]

Also, see a similar feature in total ozone in Shepherd et al. (2014), their Fig. 2a

5. One of the main conclusions of this study states (LL391-392) "*According to our results, JRA-55 is the most suitable for reanalyse trend studies due to the low significant discontinuity occurrence*". I really disagree with that. Unlike the other reanalyses, JRA-55 assimilates only total ozone observations (Kobayashi et al. 2015; Davis et al. 2017). That means that vertical distributions are constrained exclusively by an offline chemistry model. That places JRA-55 ozone in a very different category than MERRA-2 or ERA-5, both of which assimilate multiple vertically resolved data sets. The fact that no ozone profile data are assimilated in JRA-55 is consistent with the finding that there are fewer discontinuities in JRA-55 than in the other reanalyses, but it also means that any altitude-dependent trends in JRA-55 ozone, especially in the mid- to upper stratosphere are likely mainly model-generated. This is fine, of course, but I wouldn't consider them "reanalysis trends" (i.e. observation based). They're closer to model trends.

**Specific comments**

L29. Harris et al. is fine but there are a lot of more recent studies, most notably the LOTUS report (SPARC/IO3C/GAW 2019, you cite it later), and WMO 2018.

L46-47. "*the assimilation of not homogenous basic parameters*". I'm not sure what that means. "Assimilation" usually refers to assimilation of observations. What are basic parameters?

LL51-53. See my general comment 4.

LL67-101. So, this is an explanation by walking the reader through a simple example. In a way that makes it easy to understand the method but I would rather see a more general description supplemented with a discussion of assumptions, detectability thresholds, accuracy, false positives/negatives, applicability to ozone data that have trends and low-frequency modes of variability. An illustrative example is fine, but it would be good to see something more challenging than that really big discontinuity in Fig 1; something a little harder to detect, to really put the algorithm to the test. Also, is the formation of an ozone hole a relevant example? I understand that

you work with separate time series for each calendar month (40 Januaries, 40 Februaries, etc) rather than consecutive months in the calendar. Is that right?

LL124-143. Please, add details of ozone data assimilated in JRA-55 and ERA-5. Note the very different observing systems among the three reanalyses!

L139. Does ERA5 exhibit higher variability than MERRA-2 does at a given location because of its higher resolution (i.e. less averaging)? If so, how does that impact DO detection (with more "noise" it may be harder to smoke out small DOs)? How about a test where ERA5 is degraded to MERRA-2 resolution through averaging? Would that affect the results? Or perhaps monthly averages are not strongly affected by differences in resolution?

L143. The core reference for ERA5 is Hersbach et al., 2020.

LL144-145. But averaging (e.g. zonal) also reduces regional year-to-year variability, potentially bringing out large scale changes, including discontinuities, by reducing the "noise".

L152-153. The times of DO are important. Why aren't they provided?

LL154-161. See my general comment 1. Also, I think "*variance*" should be "*standard deviation*". Otherwise things wouldn't make sense dimensionally.

L236. As a way of double checking, I suggest plotting time series of differences: MERRA-2 minus JRA-55 and ERA5 minus JRA-55. Since JRA-55 doesn't assimilate anything that would directly affect ozone at 3 hPa, if there is a discontinuity in MERRA-2/ERA5 it should be evident in the difference time series. You just need to keep track of potential discontinuities in JRA-55 upper stratospheric temperature and winds that could affect ozone. Discontinuities are seen more clearly in difference time series because interannual variations (should be similar between reanalyses) get subtracted out, at least in part. Just another way of taking advantage of having several reanalyses.

LL316-321. This paragraph sounds very speculative and it's not clear to me what data it talks about: ozone, meteorological, or both? Satellite ozone observations both microwave and UV do have larger uncertainty lower down but I wouldn't say that's true for radiance data. Radiosondes are used and they do have large impacts on temperature per observation (not necessarily on ozone, though). On the other hand, ozonesondes are not assimilated by any of these reanalyses. It could be instructive to investigate more thoroughly why DOs cover larger area in the troposphere and provide a definitive answer or at least a solid hypothesis rather than speculation. Of course, I'm still not convinced that these DO profiles are accurate (general comment 1).

LL320-321. That could be tested by comparing the timings of those DO with the observing system changes.

LL326-327. Again, JRA-55 doesn't assimilate any ozone data in the upper stratosphere, so this is not that surprising.

LL332-333. "*This region is very important for vertical coupling in the middle atmosphere*". I'm not sure what this means. What coupling? Please, explain.

LL334-336. Again, what observations: ozone or radiance data?

LL338-339. I'm not sure what you mean by "the main problem".

LL342-345. "*It means there are more insignificant discontinuities in MERRA-2 than in ERA5, but the opposite is true for the significant ones. For trend analyses the significant discontinuities have larger impact on result, so the ERA5 data in the troposphere is more suitable for trend analysis than that of MERRA-2*". The first sentence implies that there are more significant DOs in ERA5 than in MERRA-2 (corroborated by Fig. 4). How does that make ERA5 more suitable for trend analyses?

LL372-373. Actually, they claim that 2004 was not important in the stratosphere except near the stratopause. Here's a copy/paste "*the new one set up in 2004, and it was strongest at 1 hPa. This maximum can be clearly explained by the transition from SBUV to EOS Aura data in 2004. **Below this layer, the flat temporal distribution of discontinuities was seen down to the troposphere, so no maximum in 2004 was observed**.*" And "*If the 2004 discontinuities were not so huge, this fact could open the possibility of using MERRA 2 ozone data in trend analysis*". This is another reason to question the algorithm: 2004 was definitely a significant discontinuity in the lower stratosphere (see Davis et al., 2017, their Fig. 11), which DO detection apparently missed.

LL389-392. See my general comment 5.

The manuscript should be edited for grammar and style if it were to be published.

**References**

Davis, S. M., et al. (2017). Assessment of upper tropospheric and stratospheric water vapor and ozone in reanalyses as part of S-RIP, Atmos. Chem. Phys., 17, 12743–12778, https://doi.org/10.5194/acp-17-12743-2017

Hersbach, H, Bell, B, Berrisford, P, et al. The ERA5 global reanalysis. *Q J R Meteorol Soc*. 2020; 146: 1999– 2049. https://doi.org/10.1002/qj.3803

Shepherd, T., Plummer, D., Scinocca, J. *et al.* Reconciliation of halogen-induced ozone loss with the total-column ozone record. *Nature Geosci* **7,** 443–449 (2014). https://doi.org/10.1038/ngeo2155

Wargan, K., G. Labow, S. Frith, S. Pawson, N. Livesey, and G. Partyka, 2017: Evaluation of the Ozone Fields in NASA's MERRA-2 Reanalysis. *J. Climate*, **30**, 2961–2988, https://doi.org/10.1175/JCLI-D-16-0699.1.

WMO (World Meteorological Organization) (2018). Scientific Assessment of Ozone Depletion: 2018. *Global Ozone Research and Monitoring Project*–Report No. 58, 588 pp., Geneva, Switzerland.

---

## Referee Comment (RC2) · Anonymous Referee #2 · 21 Sep 2020

Review of „Occurrence of discontinuities in the ozone concentration data from three reanalyses" by P. Krizan, M. Kozubek, and J. Lastovicka.

The manuscript objective is to search for step changes in the time series of ozone profiles, between 500 hPa and 1 hPa, from three reanalyses, MERRA-2, Era-2, and JRA-55. If the step changes are spurious (i.e. not related to the atmosphere processes) and enough large the trend estimations will be unreliable as forced by changes in technical details of the reanalysis method (e.g. inclusion new satellite data and/or procedure in GCM). Therefore, the subject is important and fits perfectly to the aim of the ACP journal. However, **the manuscript in present form is not ready for publication in the journal and should be rejected**. It requires substantial changes prior any submission.

*General Comments*.

The authors use the Pettitt test to detect inhomogeneities in the ozone time series. They do not provide reasons for choosing this test and do not discuss its applicability to the ozone time series. There are many other tests to detect series homogeneity (see C. Yozgatligil and C. Yazici, https://doi.org/10.1002/joc.4329). Here, the Pettitt test is used rather mechanically assuming only one change point. The authors are aware that this is not the case for the analysed ozone time series as they discussed possible presence of two change points (~2003 and ~2015, l.369-373). The test works well for cases with a singular discontinuity close to the centre of the series. Below, there is an example illustrating that the Pettitt test fails when multiple change points are present in time series.

[Figure]

Here, the Pettitt test is applied to artificial time series with two change points. 10 values (points in red) are added to the authors' time series (Fig.1 in the manuscript) discussed at the beginning of Section 2 (l.65-101). In this way, a downward jump (at 21$^{th}$ point of the series) to the time series value at 10$^{th}$ point is modelled. P value, which is found at the point with maximum U (12$^{th}$ point according Eq.1), is equal to ~0.15 (according Eqs. 2-3) i.e. above 0.05 limit. This allows to formulate a hypothesis about the lack of discontinuities in extended time series but

the authors' original series (20 blue points) showed clear discontinuity at 11th point with P=0.0092. Thus, the assumption of only one changing point in the analysed time series is crucial for this test performance. Therefore, it seems that the test should be repeatedly performed for the connected parts of the time series, not just once for the whole series.

To make the problem even more difficult, there are possible change points in the ozone time series due to superposition of "natural" ozone forcing factors (QBO, ENSO, Brewer Dobson circulation, the Arctic Oscillations, persistence of lack of sudden stratospheric warming in some periods, etc.). Methods of distinguishing between false and "natural" points of change should at least be discussed in the manuscript.

In the reviewer opinion, an analysis of the change point time is necessary. Just calculation discontinuity occurrences over globe is not enough. History of changes in the reanalyses' methodology has been known and these changes should be linked with the time of step changes disclosed in the time series.

The authors define two types of discontinuity: insignificant and significant. They claimed that only significant ones can erroneously affect the anthropogenic trend values as opposed to the insignificant ones. This suggestion needs justification and should be applied only to spurious discontinuities if they are correctly selected from the ozone time series.

Taking into account all mentioned above problems, the conclusions (especially the last one) are very doubtful.

*Specific Comments*

It is not clear how the significant differences between in the discontinuity occurrence in the reanalyses are calculated. At first the authors claim (l.193) that "All differences above 1% in absolute value must be regarded as significant because the number of grids is very high (1038240 for ERA5 and 207936 for MERRA-2 ". So, practically all differences shown in Tab.3 are significant. But a few lines later (l.197-199) they state "The variance of DO is at some layers high, so it is the reason why the differences between MERRA-2 and ERA5 are insignificant at the majority of layers". Something is wrong. Please describe the test used to find significance of the differences between DO by different reanalyses. Number of independent cases (i.e. degrees of freedom) is usually used in the calculation of the test significance, not number of all data points, because the observations at neighbouring points are usually highly correlated. A calculation of number of independent data points is not a simple task and depends on the spatial correlation structure of the data.

The reviewer found a problem to understand vertical profiles of DO. I guess (the authors do not provide explanation) that extreme (minimal or maximal) DO in Fig.2 (and in many others Figures) at selected level is shown for the specific month, and average DO is the mean from 12 monthly values. If this is OK why there are so large monthly variations in DO for the fixed layer (it is seen as large distance between max and min profiles, Fig.2). Spurious change step linked with changes in reanalysis methodology should appear simultaneously in all months. Large intra year variability of DO suggests that step changes may include a kind of mixing between "natural" (dynamically driven in dependence of season of the year) and spurious step changes.

The authors define significant step changes in the data using 1-sigma criterion of the difference between the mean values before and after the jump. Here, the reviewer does not discuss if this threshold is enough large to affect the trend calculation. Different problem is how localization of this jump affects trend calculation. It seems that the effect will be strongest when the jump occurs in the middle of the time series. Thus, not only the difference between the means is import here. Presence of multiple step changes affects the mean value after (or before) the jump, so significance of the step change should be calculated taking into account the mean derived from the period between the step changes (e.g. period between 11[th] and 20[th] point in the attached Figure). Therefore, the selection of significant step changes needs at least discussion in the manuscript. Searing for a link between spurious step changes and trend calculations requires much more efforts (maybe in new manuscript?) and any statement concerning it should be only hypothesized (and omitted from conclusions) in the present manuscript.

The authors use formula for one-sided probability (Eq. 2, line 87) in the illustration of the Pettitt test. It should be two times larger for two-sided probability, i.e. $P=2\exp(T)$, if the direction of change (up or down) after the step change is not important. Please check if P is used correctly in the rest part of the manuscript.